# Design, Synthesis, and Bioactivities of Novel Tryptophan Derivatives Containing 2,5-Diketopiperazine and Acyl Hydrazine Moieties

**DOI:** 10.3390/molecules27185758

**Published:** 2022-09-06

**Authors:** Lili Li, Rongxin Yang, Jianhua Liu, Jingjing Zhang, Hongjian Song, Yuxiu Liu, Qingmin Wang

**Affiliations:** 1China State Key Laboratory of Elemento-Organic Chemistry, College of Chemistry, Frontiers Science Center for New Organic Matter, Nankai University, Tianjin 300071, China; 2College of Basic Science, Tianjin Agricultural University, Tianjin 300384, China

**Keywords:** tryptophan, DKP, acylhydrazone, anti-TMV activity, fungicidal activity, larvicidal activity

## Abstract

Based on the scaffolds widely used in drug design, a series of novel tryptophan derivatives containing 2,5-diketopiperazine and acyl hydrazine moieties have been designed, synthesized, characterized, and evaluated for their biological activities. The bioassay results showed that the target compounds possessed moderate to good antiviral activities against tobacco mosaic virus (TMV), among which compounds **4**, **9**, **14**, **19**, and **24** showed higher inactivation, curative, and protection activities in vivo than that of ribavirin (39 ± 1, 37 ± 1, 39 ± 1 at 500 mg/L) and comparable to that of ningnanmycin (58 ± 1, 55 ± 1, 57 ± 1% at 500 mg/L). Thus, these compounds are a promising candidate for anti-TMV development. Most of these compounds showed broad-spectrum fungicidal activities against 13 kinds of phytopathogenic fungi and selective fungicidal activities against *Alternaria solani*, *Phytophthora capsica*, and *Sclerotinia sclerotiorum*. Additionally, some of these compounds exhibited larvicidal activities against *Tetranychus cinnabarinus*, *Plutella xylostella*, *Culex pipiens pallens*, *Mythimna separata*, *Helicoverpa armigera*, and *Pyrausta nubilalis*.

## 1. Introduction

Plant viruses, which are composed of nucleic acids and proteins [1], cause global economic losses as high as USD 60 billion every year [2,3,4,5,6]. They can change the normal metabolic process of host plants, interfere with or destroy the activity of respiratory photosynthetic enzymes and the metabolism of auxin and other hormones, in addition to robbing some nutrients of infected plants. Thus far, about 1100 kinds of viruses have been found. TMV (tobacco mosaic virus) is one of the oldest known plant viruses and ranks first among the top 10 plant viruses, causing economic losses of more than USD 100 million per year. There is no antiviral agent that can completely inhibit plant viruses, and the development of novel and more practical antiviral reagents is sorely needed [7,8].

Natural products are secondary metabolites retained by natural selection after a long time of evolution. Natural products are often characterized by chemical structure and biological activity diversity, which makes them of great value in drug development and utilization [9,10]. By September 2019, among the 185 small molecule anticancer drugs approved for sale by the FDA, 120 are related to natural products, accounting for 64.9% [11].

Tryptophan is a biosynthetic precursor in notable bioactive compounds [12,13,14,15], it also has a central role in metabolism, protein structure, and signaling, and analogs are frequently used to probe enzyme function or alter enzyme properties. In our previous work, we found, for the first time, that tryptophan showed moderate anti-plant virus activity [16], which can be used as an antiviral lead for subsequent studies.

2,5-Diketopiperazines (DKP) occur in a variety of natural products from bacteria, fungi, the plant kingdom, and mammals (Figure 1) [17,18]. They are not only a class of natural privileged structures that can bind to a wide range of receptors, but they also have several advantages, such as constrained conformation, are chirally enriched, stable to proteolysis, and can mimic a preferential peptide conformation, which makes them attractive scaffolds for drug discovery [19,20].

The acyl hydrazone structure is a complex of hydrogen bond donors and receptors. In our previous work, it was found that the acyl hydrazone structure could enhance the anti-TMV activity of the compound, possibly because the hydrogen bond receptor or donor of the acyl hydrazone enhanced the interaction with the amino acid residues of TMV CP, thus preventing the assembly of the virus [21,22,23].

In this work, to improve the anti-virus activity of tryptophan, we designed and synthesized a series of novel tryptophan derivatives containing diketopiperazine (DKP) and acyl hydrazon moieties and first evaluated their biological activities (Figure 2). In addition, the fungicidal and larvicidal activities of the newly synthesized tryptophan derivatives were also studied to expand their potential agricultural applications.

## 2. Results and Discussion

### 2.1. Synthesis

Using natural amino acid L-tryptophan as raw material, through esterification, amidation, cyclization, and condensation reactions, we could easily realize the synthesis of target compounds **3**–**32** (Figure 1) [23]. Compared with compound **3**, the notable feature of the ^1^H NMR spectrum of these target compounds was an additional single peak of imine hydrogen (See the Appendix A for details). As a key step, a rigid diketopiperazine ring was obtained by the microwave-assisted hydrazinolysis reaction with hydrazine hydrate. Compared with conventional heating reactions, the efficiency and yield of the microwave reaction were improved (Table 1).

### 2.2. Biological Assay

#### 2.2.1. Anti-TMV Activities

Using the commercial plant viricides ningnanmycin and ribavirin as controls, we first evaluated the inactivation effect of synthetic derivatives **3**–**32** against TMV in vivo at 500 mg/L, and then the curative and protective modes of antiviral activity were tested at both 500 and 100 mg/L for these derivatives with more than 40% in vivo inactivation effect at 500 mg/L. The bioactive results in Table 2 show that most of these derivatives exhibited better antiviral activity than L-tryptophan. The introduction of the acylhydrazone structure was beneficial to the antiviral activity of these compounds; most of these derivatives **4**–**32** exhibited better antiviral activity than acylhydrazine derivative, **3**, which indicated that the acylhydrazone moiety played an important role in improving the antiviral activity. The derivatives containing the structure of benzyl imines (**4**–**25**) exhibited better antiviral activity than that of heteroarylmethyl imines (**26**–**30**) and alkyl imines (**31**, **32**).

For acylhydrazone derivatives, **4**–**25**, the types, position, and number of substituents on the benzene ring had an important influence on the anti-TMV activity. The introduction of strong electron-withdrawing groups on the benzene ring, such as nitro (**5**, **17**), and trifluoromethyl (**10**), was detrimental to the activity. For the substituents at the para position of the benzene ring, electron-donating groups (**6**, **9**) and weak electron-withdrawing group (**8**) were favorable for maintaining the activity. The position of the substituents on the benzene ring had a significant effect on the activity and showed a significant ortho-position effect; that is, the activities of the ortho-substituted derivatives were significantly better than that of the derivatives substituted at other positions (**14** versus **8**, **9**, and **19** versus **9**, **18**). For example, when the benzene ring has a methoxy substituted on the benzene ring, the order of bioactivity levels is **19** (2-OMe) > **9** (4-OMe) > **18** (3-OMe); different from this, when the substituent was chlorine, the order changed to **14** (2-Cl) > **13** (3-Cl) > **8** (4-Cl). The anti-TMV activities of **14** (inhibition rate for inactivation, curative, and protection activities in vivo: 54 ± 3, 50 ± 3, 45 ± 2% at 500 mg/) and **19** (53 ± 2, 48 ± 4, 45 ± 2% at 500 mg/L) were better than that of ribavirin (39 ± 1, 37 ± 1, 39 ± 1 at 500 mg/L) and comparable to that of ningnanmycin (58 ± 1, 55 ± 1, 57 ± 1% at 500 mg/L). These two compounds could be further developed as antiviral drug candidates.

Moreover, the number of substituents on the benzene ring affected the activity. Increasing the number of substituents was not beneficial to improving the activity of these derivatives, such as, **8** (4-Cl, inhibition rate for inactivation, curative, and protection activities in vivo: 42 ± 1, 48 ± 4, 39 ± 3% at 500 mg/L), **13** (3-Cl, 47 ± 1, 49 ± 4, 42 ± 4% at 500 mg/L), and **14** (2-Cl, 54 ± 3, 50 ± 3, 45 ± 2% at 500 mg/L) versus **15** (2,4-diCl) (42 ± 1, 35 ± 4, 32 ± 2% at 500 mg/L) and **16** (3,4-diCl) (24 ± 4% at 500 mg/L). It was interesting that **21** (42 ± 3, 44 ± 3, 39 ± 2% at 500 mg/L) and **22** (49 ± 4, 46 ± 2, 50 ± 3% at 500 mg/L) displayed better activities than **20** (37 ± 3% at 500 mg/L). We speculated that the existence of ring tension was beneficial to improving the activity.

For heterocyclic aromatic compounds, they showed the following order of bioactivity levels **29** (imidazolyl, inhibition rate for inactivation, curative, and protection activities in vivo: 43 ± 2, 41 ± 3, 46 ± 2%, 500 mg/L) > **26** (pyrrolyl, 39 ± 3, 35 ± 3, 46 ± 1%, 500 mg/L) > **28** (furyl, 37 ± 2%, 500 mg/L) ≈ **27** (thienyl, 33 ± 4%, 500 mg/L) > **30** (pyridyl, 31 ± 1%, 500 mg/L).

To investigate the role of R′ in bioactivity, we designed and synthesized compound **25**, which has a methyl at the imine moiety. To our delight, it showed lower antiviral activities (43 ± 3, 38 ± 2, 40 ± 4%, 500 mg/L) than compound **4** (R′ = H, 51 ± 1, 46 ± 2, 48 ± 3% at 500 μg/mL). The above experimental results prove the rationality of our choice of aldimine. When the benzene ring was changed to alkyl groups (**31** and **32**), the activity decreased obviously.

#### 2.2.2. Fungicidal Activities

Fungicidal activities were evaluated by the mycelial growth method. In general, most derivatives exhibited a broad spectrum of fungicidal activities against 13 kinds of phytopathogenic fungi (Table 3). The fungicidal spectrum of most acylhydrazone derivatives (**4**–**32**) was broader than compound **3**, and their fungicidal activities were also higher than compound **3**. Almost all these compounds showed fungicidal activities selectively against *Alternaria solani*, *Phytophthora capsica*, and *Sclerotinia sclerotiorum*. Among them, compounds **5** (4-nitrobenzyl imine), **9** (4-methoxybenzyl imine), **13** (3-clorobenzyl imine), **15** (2,4-dichlorobenzyl imine), **19** (2-clorobenzyl imine), and **24** (3,5-di-tert-butyl-4-hydroxybenzyl imine) showed > 50% fungicidal activities against more than five kinds of fungi. Compounds **15** and **24** had a more broad-spectrum fungicidal activity and showed more than 50% fungicidal activities against 9 fungi and 12 fungi, respectively. Compounds **15** and **24** exhibited > 90% against *Phytophthora capsica* at 50 mg/L, and compound **15** showed > 90% fungicidal activity against *Sclerotinia sclerotiorum*, *Botrytis cinerea Pers.ex Fr.*, and *Rhizoctonia solani* at 50 mg/L, specifically. Interestingly, compound **16** (3,4-dichlorobenzyl imine), which had a similar structure to the derivative **15**, did not show good fungicidal activities. The only difference between these two derivatives was the positions of the substituents, which indicated that the substituent on the benzene ring also had an important influence on the fungicidal activity.

#### 2.2.3. Larvicidal Activities

We then studied the larvicidal activities of the synthesized derivatives, and different orders of pests were selected for the research, such as *T. cinnabarinus*, *P. xylostella* (lepidoptera), and *C. pipiens pallens* (diptera) (Table 4). In general, some derivatives showed larvicidal activities against these pests, and at the same time, these derivatives showed obvious selectivity. The derivatives containing the structure of benzyl imines **18** (3-OMe) and **21** (1,3-dioxol) showed good larvicidal activity against *T. cinnabarinus*. Hydrazide derivative **3** showed no activity against *T. cinnabarinus*. For the lepidopteran pest *P. xylostella*, the overall activity was better than that against *T. cinnabarinus*, and most of the derivatives showed larvicidal activities. Likewise, hydrazide derivative **3** did not exhibit larvicidal activity against *P. xylostella*. Derivatives containing the structure of benzyl imines **4** (no substituent), **23** (4-bromo-2,6-difluoro), and heteroarylmethyl imines **29** (imidazolyl) showed >50% larvicidal activities against *P. xylostella* at 200 mg/L. Different from the activity rules of the former two pests, hydrazide derivative **3** has larvicidal activity against *C. pipiens pallens*, and its activity against *C. pipiens pallens* larvae was 50 ± 0% at the concentration of 2 mg/L. Derivatives containing the structure of benzyl imines **9** (4-OMe), **21** (1,3-dioxol), **23** (4-bromo-2,6-difluoro), and heteroarylmethyl imines **28** (furyl) showed >60% larvicidal activities at 5 mg/L.

To further study the larvicidal activities of these derivatives against other lepidopteran pests, the larvicidal activities against *M. separate*, *H. armigera*, and *P. nubilalis* were also studied (Table 5). In general, most derivatives showed larvicidal activities against these three lepidopteran pests. The structure–activity relationship was different from that of larvicidal activities against *P. xylostella*, where derivative **3** showed no larvicidal activity, but this derivative exhibited larvicidal activity against these three lepidopteran pests. Derivatives containing the structure of benzyl imines **12** (4-Ph), heteroarylmethyl imines **28** (furyl), alkyl imines **31** (*t*-Bu), and **32** (cyclohexyl) showed >60% larvicidal activities at 600 mg/L, the larvicidal activities of derivatives **31** and **32** against these three pests were 100% at 600 mg/L. This means that a good fat-soluble alkyl substituent was beneficial to larvicidal activities. Derivatives **31** and **32** can be used as insecticidal leads for further study.

## 3. Materials and Methods

### 3.1. Materials

The hydrazinolysis reaction was carried out in a microwave synthesis system (100 °C, 100 W, Discover S-Class, CEM). ^1^H, ^13^C nuclear magnetic resonance (NMR) spectra were obtained at 400 MHz using a Bruker AC-P 400. Chemical shift values (*δ*) were given in parts per million (ppm) and were downfield from internal tetramethylsilane. High-resolution mass spectra (HRMS) data were obtained on an FTICR-MS instrument (Ionspec 7.0 T). The melting points were determined on an X-4 binocular microscope melting point apparatus and were uncorrected. Reaction progress was monitored by thin-layer chromatography on silica gel GF-254 with detection by UV.

Ribavirin (Topscience Co., Hongkong, China), ningnanmycin (Alta Scientific Co., Tianjin, China), chlorothalonil (Bailing Agrochemical Co., Jiangsu, China), rotenone (Accela ChemBio Inc., Shanghai, China), avermectin (Chemieliva Pharmaceutical Co., Chongqing, China), and other reagents were purchased from commercial sources and were used as received.

### 3.2. General Synthesis

The synthetic routes of target compounds **3**–**32** are depicted in Figure 1. The spectra of target compounds **3**–**32** are depicted in the Appendix A.

#### 3.2.1. Synthesis of (S)-Methyl 2-Amino-3-(1H-indol-3-yl)propanoate (**1**)

To a solution of L-tryptophan (10 g, 48.97 mmol) in anhydrous methanol (150 mL) SOCl_2_ (10 mL) was added dropwise slowly and then heated at 100 °C. When the reaction was complete, as indicated by thin-layer chromatography (5 h), the reaction mixture was cooled to room temperature, then the mixture was concentrated in vacuo and washed with anhydrous Na_2_CO_3_ saturated solution, extracted with ethyl acetate (50 mL × 3), and the combined organic phases were washed with brine, dried over Na_2_SO_4_, and filtered; the filtrate was evaporated under reduced pressure to give a brown solid (9.71 g, 91%, mp 90–91 °C). ^1^H NMR (400 MHz, CDCl_3_) *δ* 8.35 (s, 1H, Ar-N**H**), 7.61 (d, *J* = 7.6 Hz, 1H, Ar-H), 7.33 (d, *J* = 8.0 Hz, 1H, Ar-H), 7.19 (t, *J* = 7.6 Hz, 1H, Ar-H), 7.12 (t, *J* = 7.6 Hz, 1H, Ar-H), 7.02 (d, *J* = 2.0 Hz, 1H, Ar-H), 3.84 (dd, *J* = 7.6, 4.8 Hz, 1H, C**H**-NH_2_), 3.71 (s, 3H, OC**H_3_**), 3.28 (dd, *J* = 14.4, 4.8 Hz, 1H, C**H_2_**-CH), 3.05 (dd, *J* = 14.4, 8.0 Hz, 1H, C**H_2_**-CH), 1.64 (s, 2H, N**H_2_**). ^13^C NMR (100 MHz, CDCl_3_) *δ* 175.8, 136.3, 127.5, 123.0, 122.2, 119.5, 118.8, 111.3, 111.0, 55.0, 52.1, 30.8.

#### 3.2.2. Synthesis of (S)-Methyl 2-(2-Chloroacetamido)-3-(1H-indol-3-yl)propanoate (**2**)

A mixture of **1** (19.61 g, 44.0 mmol) and NaHCO_3_ (5.6 g, 66.0 mmol) in dichloromethane (150 mL) was cooled in an ice bath, chloroacetyl chloride (5.0 mL) was added dropwise slowly, and the reaction mixture was allowed to warm to room temperature and then continuously stirred for 14 h. Then the mixture was quenched with an anhydrous NaHCO_3_ solution and then extracted with dichloromethane (50 mL × 3); the combined organic phases were washed with brine (50 mL), dried over anhydrous Na_2_SO_4_, filtered, and concentrated in vacuo to afford **2** as a brown liquid (11.77 g, 91%). ^1^H NMR (400 MHz, CDCl_3_) *δ* 8.43 (s, 1H), 7.52 (d, *J* = 8.0 Hz, 1H, Ar-H), 7.32 (d, *J* = 8.0 Hz, 1H, Ar-H), 7.18 (t, *J* = 7.2 Hz, 1H, Ar-H), 7.10 (m, 2H, Ar-H, C=C**H**-NH), 6.97 (s, 1H), 4.90 (q, 1H, C**H**-NH), 3.96 (s, 2H, C**H**_2_-Cl), 3.68 (s, 3H, -OC**H_3_**), 3.34 (d, *J* = 5.6 Hz, 2H, C**H**_2_-CH). ^13^C NMR (100 MHz, CDCl_3_) *δ* 171.7, 165.8, 136.2, 127.4, 122.8, 122.4, 119.8, 118.5, 111.4, 109.6, 53.2, 52.6, 42.5, 27.5.

#### 3.2.3. Synthesis of (S)-3-((1H-Indol-3-yl)methyl)-1-aminopiperazine-2,5-dione (**3**)

A mixture of **2** (1.18 g, 4.00 mmol) and hydrazine hydrate (80%) (2.5 equiv, 10.00 mmol) in ethanol (20 mL) was in a microwave vessel and heated by microwaves (100 W, 100 °C) for 15 min. Then the mixture was naturally cooled to room temperature, filtered, washed with a small amount of alcohol, dried, and gave **3** as a white solid (0.97 g, 95%, mp 211–212 °C). ^1^H NMR (400 MHz, DMSO*-d_6_*) *δ* 10.95 (s, 1H, Ar-N**H**), 8.25 (s, 1H, CH-N**H**), 7.47 (d, *J* = 8.0 Hz, 1H, Ar-H), 7.33 (d, *J* = 8.0 Hz, 1H, Ar-H), 7.05 (t, *J* = 7.6 Hz, 2H, Ar-H), 6.96 (t, *J* = 7.2 Hz, 1H, Ar-H), 4.73 (s, 2H, N**H_2_**), 4.54–3.96 (m, 1H, C**H**-NH), 3.47 (d, *J* = 17.2 Hz, 1H, CH_2_-C=O), 3.25 (dd, *J* = 14.4, 4.0 Hz, 1H, C**H**_2_-CH), 3.01 (dd, *J* = 14.4, 4.0 Hz, 1H, C**H**_2_-CH), 2.77 (d, *J* = 17.2 Hz, 1H, C**H**_2_-C=O). ^13^C NMR (100 MHz, DMSO*-d_6_*) *δ* 165.2, 164.7, 136.4, 127.9, 125.2, 121.4, 119.0, 118.9, 111.7, 108.3, 55.4, 52.6, 30.2. HRMS (ESI) calcd for C_13_H_14_N_4_O_2_(M+H)^+^ 259.1195, found 259.1187.

#### 3.2.4. General Synthesis Route for Derivatives **4**–**32**

To a solution of **3** (0.4 g, 1.549 mmol) in ethanol (30 mL), aldehyde (1.2 equiv) was added, then the mixture was heated at 100 °C for 12 h and filtered or purified by silica column chromatography (dichloromethane: methanol = 10:1) to afford **4**–**32**.

(*S*)-3-((1*H*-indol-3-yl)methyl)-1-(benzylideneamino)piperazine-2,5-dione (**4**). White solid, 0.49 g, 91%, mp 290-296 °C. ^1^H NMR (400 MHz, DMSO*-d_6_*) *δ* 10.98 (s, 1H, Ar-N**H**), 8.51 (s, 1H, N**H**-CH), 7.89 (s, 1H, Ar-C**H=**N), 7.68 (d, *J* = 4.4 Hz, 2H, Ar-H), 7.49 (d, *J* = 7.6 Hz, 1H, Ar-H), 7.44 (m, 3H, Ar-H), 7.31 (d, *J* = 8.0 Hz, 1H, Ar-H), 7.05 (s, 1H, C=C**H**-NH), 7.00 (t, *J* = 7.6 Hz, 1H, Ar-H), 6.89 (t, *J* = 7.6 Hz, 1H, Ar-H), 4.35 (s, 1H, C**H**-NH), 3.99 (d, *J* = 16.8 Hz, 1H, C**H**_2_-C=O), 3.34 (m, 1H, C**H**_2_-CH), 3.19–3.03 (m, 2H, C**H**_2_-CH, C**H**_2_-C=O). ^13^C NMR (100 MHz, DMSO*-d_6_*) *δ* 164.2, 163.9, 147.6, 136.4, 134.7, 130.8, 129.2, 128.0, 127.8, 125.3, 121.4, 119.2, 118.9, 111.7, 108.3, 56.3, 49.2, 30.8. HRMS (ESI) calcd for C_20_H_18_N_4_O_2_(M+H)^+^ 347.1508, found 347.1504.

(*S*)-3-((1*H*-indol-3-yl)methyl)-1-(4-nitrobenzylideneamino)piperazine-2,5-dione (**5**). Yellow solid, 0.41 g, 91%, mp 252–253 °C. ^1^H NMR (400 MHz, DMSO*-d_6_*) *δ* 10.98 (s, 1H, Ar-N**H**), 8.55 (s, 1H, N**H**-CH), 8.30 (d, *J* = 8.8 Hz, 2H, Ar-H, Ar-C**H=**N), 8.00–7.86 (m, 3H, Ar-H), 7.47 (d, *J* = 7.6 Hz, 1H, Ar-H), 7.30 (d, *J* = 8.0 Hz, 1H, Ar-H), 7.06 (s, 1H, C=C**H**-NH), 6.98 (t, *J* = 7.6 Hz, 1H, Ar-H), 6.87 (t, *J* = 7.6 Hz, 1H, Ar-H), 4.40–4.39 (m, 1H, C**H**-NH), 4.03 (d, *J* = 16.8 Hz, 1H, C**H**_2_-C=O), 3.35 (dd, *J* = 14.8, 4.4 Hz, 1H, C**H**_2_-CH), 3.13 (dd, *J* = 14.8, 4.4 Hz, 1H, C**H**_2_-CH), 3.07 (d, *J* = 16.8 Hz, 1H, C**H**_2_-C=O). ^13^C NMR (100 MHz, DMSO*-d_6_*) *δ* 164.3, 163.9, 148.4, 143.7, 141.1, 136.4, 128.8, 127.7, 125.4, 124.5, 121.4, 119.2, 118.9, 111.7, 108.1, 56.3, 49.1, 30.9. HRMS (ESI) calcd for C_20_H_17_N_5_O_4_ (M+H)^+^ 392.1359, found 392.1351.

(*S*)-3-((1*H*-indol-3-yl)methyl)-1-(4-tert-butylbenzylideneamino)piperazine-2,5-dione (**6**). White solid, 0.44 g, 94%, mp 316–317 °C. ^1^H NMR (400 MHz, DMSO*-d_6_*) *δ* 10.95 (s, 1H, Ar-N**H**), 8.46 (d, *J* = 2.4 Hz, 1H, N**H**-CH), 7.89 (s, 1H, Ar-C**H=**N), 7.60 (d, *J* = 8.4 Hz, 2H, Ar-H), 7.47 (m, 3H, Ar-H), 7.31 (d, *J* = 8.0 Hz, 1H, Ar-H), 7.05 (d, *J* = 2.0 Hz, 1H, C=C**H**-NH), 7.01 (t, *J* = 8.0 Hz, 1H, Ar-H), 6.89 (t, *J* = 7.6 Hz, 1H, Ar-H), 4.45–4.25 (m, 1H, C**H**-NH), 3.96 (d, *J* = 16.8 Hz, 1H, C**H**_2_-C=O), 3.37–3.31 (m, 1H, C**H**_2_-CH), 3.20–3.04 (m, 2H, C**H**_2_-CH, C**H**_2_-C=O), 1.29 (s, 9H, C(C**H_3_**)_3_). ^13^C NMR (100 MHz, DMSO*-d_6_*) *δ* 164.3, 163.8, 153.7, 148.1, 136.4, 132.0, 127.8, 127.8, 126.0, 125.3, 121.4, 119.2, 118.9, 111.7, 108.3, 56.3, 49.3, 35.1, 31.4, 30.7.HRMS (ESI) calcd for C_24_H_26_N_4_O_2_(M+H)^+^ 403.2134, found403.2123.

(*S*)-3-((1*H*-indol-3-yl)methyl)-1-(4-(dimethylamino)benzylideneamino)piperazine-2,5-dione (**7**). White solid, 0.85 g, 96%, mp 284–285 °C. ^1^H NMR (400 MHz, DMSO*-d_6_*) *δ* 10.96 (s, 1H, Ar-N**H**), 8.43 (d, *J* = 2.4 Hz, 1H, N**H**-CH), 7.80 (s, 1H, Ar-C**H=**N), 7.52–7.46 (m, 3H, Ar-H), 7.32 (d, *J* = 8.0 Hz, 1H, Ar-H), 7.06 (d, *J* = 2.0 Hz, 1H, C=C**H**-NH), 7.02 (t, *J* = 7.6 Hz, 1H, Ar-H), 6.91 (t, *J* = 7.2 Hz, 1H, Ar-H), 6.72 (d, *J* = 8.8 Hz, 2H, Ar-H), 4.32–4.24 (m, 1H, C**H**-NH), 3.89 (d, *J* = 16.8 Hz, 1H, C**H**_2_-C=O), 3.33–3.29 (m, 1H, C**H**_2_-CH), 3.15–3.06 (m, 2H, C**H**_2_-CH, C**H**_2_-C=O), 2.96 (s, 6H, N(C**H**_3_)_2_). ^13^C NMR (100 MHz, DMSO*-d_6_*) *δ* 164.0, 162.7, 151.8, 151.2, 135.9, 129.0, 127.4, 124.7, 121.1, 120.9, 118.8, 118.4, 111.5, 111.2, 107.9, 55.8, 49.3, 39.7, 30.1. HRMS (ESI) C_22_H_23_N_5_O_2_ calcd for (M+H)^+^ 390.1930, found 390.1926.

(*S*)-3-((1*H*-indol-3-yl)methyl)-1-(4-chlorobenzylideneamino)piperazine-2,5-dione (**8**). White solid, 0.67 g, 91%, mp 279–280 °C. ^1^H NMR (400 MHz, DMSO*-d_6_*) *δ* 10.96 (s, 1H, Ar-N**H**), 8.49 (d, *J* = 2.4 Hz, 1H, N**H**-CH), 7.88 (s, 1H, Ar-C**H=**N), 7.69 (d, *J* = 8.4 Hz, 2H), 7.51 (d, *J* = 8.8 Hz, 2H), 7.47 (d, *J* = 8.0 Hz, 1H, Ar-H), 7.30 (d, *J* = 8.0 Hz, 1H, Ar-H), 7.04 (d, *J* = 2.4 Hz, 1H, C=C**H**-NH), 6.99 (t, *J* = 7.6 Hz, 1H, Ar-H), 6.88 (t, *J* = 7.6 Hz, 1H, Ar-H), 4.39– 4.31(m, 1H, C**H**-NH), 3.97 (d, *J* = 16.8 Hz, 1H, C**H**_2_-C=O), 3.36 (dd, *J* = 14.4, 4.0 Hz, 1H, C**H**_2_-CH), 3.14 –3.05 (m, 2H, C**H**_2_-CH, C**H**_2_-C=O). ^13^C NMR (100 MHz, DMSO*-d_6_*) *δ* 163.6, 163.4, 145.2, 135.9, 134.7, 133.2, 129.0, 128.8, 127.2, 124.8, 120.9, 118.7, 118.4, 111.2, 107.7, 55.8, 48.6, 30.3. HRMS (ESI)C_20_H_17_ClN_4_O_2_ calcd for (M+H)^+^ 381.1118, found 381.1107.

(*S*)-3-((1H-indol-3-yl)methyl)-1-(4-methoxybenzylideneamino)piperazine-2,5-dione (**9**). White solid, 0.97 g, 93%, mp 287–288 °C. ^1^H NMR (400 MHz, DMSO*-d_6_*) *δ* 10.96 (s, 1H, Ar-N**H**), 8.46 (d, *J* = 2.8 Hz, 1H, N**H**-CH), 7.87 (s, 1H, Ar-C**H=**N), 7.62 (d, *J* = 8.8 Hz, 2H, Ar-H), 7.49 (d, *J* = 8.0 Hz, 1HAr-H), 7.31 (d, *J* = 8.0 Hz, 1H, Ar-H), 7.05 (d, *J* = 2.4 Hz, 1H, C=C**H**-NH), 7.03–6.98 (m, 3H, Ar-H), 6.89 (t, *J* = 7.2 Hz, 1H, Ar-H), 4.36–4.28 (m, 1H, C**H**-NH), 3.94 (d, *J* = 16.8 Hz, 1H, C**H**_2_-C=O), 3.80 (s, 3H, -OC**H_3_**), 3.38–3.31 (m, 1H, C**H**_2_-CH), 3.15–3.04 (m, 2H, C**H**_2_-CH, C**H**_2_-C=O). ^13^C NMR (100 MHz, DMSO*-d_6_*) *δ* 163.8, 163.1, 161.0, 148.1, 135.9, 129.1, 127.3, 126.7, 124.8, 120.9, 118.8, 118.4, 114.2, 111.2, 107.8, 55.8, 55.3, 48.9, 30.2.HRMS (ESI) C_21_H_20_N_4_O_3_calcd for (M+H)^+^ 377.1613, found 377.1608.

(*S*)-3-((1*H*-indol-3-yl)methyl)-1-(4-(trifluoromethyl)benzylideneamino)piperazine-2,5-dione (**10**). White solid, 0.43 g, 91%, mp 278–279 °C. ^1^H NMR (400 MHz, DMSO*-d_6_*) *δ* 10.97 (s, 1H, Ar-N**H**), 8.52 (s, 1H, N**H**-CH), 7.94 (s, 1H, Ar-C**H=**N), 7.88 (d, *J* = 8.0 Hz, 2H, Ar-H), 7.81 (d, *J* = 8.4 Hz, 2H, Ar-H), 7.47 (d, *J* = 8.0 Hz, 1H, Ar-H), 7.30 (d, *J* = 8.0 Hz, 1H, Ar-H), 7.05 (s, 1H, C=C**H**-NH), 6.98 (t, *J* = 7.2 Hz, 1H, Ar-H), 6.87 (t, *J* = 7.6 Hz, 1H, Ar-H), 4.37 (s, 1H, C**H**-NH), 4.02 (d, *J* = 16.8 Hz, 1H, C**H**_2_-C=O), 3.41–3.35 (m, 1H, C**H**_2_-CH), 3.19–3.03 (m, 2H, C**H**_2_-CH, C**H**_2_-C=O). ^13^C NMR (100 MHz, DMSO*-d_6_*) *δ* 163.7, 163.5, 144.3, 138.3, 135.9, 128.0, 127.2, 125.7, 124.9, 120.9, 118.7, 118.4, 55.8, 48.6, 40.2, 30.4.HRMS (ESI) C_21_H_17_F_3_N_4_O_2_calcd for (M+H)^+^ 415.1382, found 415.1377.

(*S*)-4-((3-((1*H*-indol-3-yl)methyl)-2,5-dioxopiperazin-1-ylimino)methyl)benzonitrile (**11**). White solid, 0.52 g, 90%, mp 249–250 °C. ^1^H NMR (400 MHz, DMSO*-d_6_*) *δ* 10.97 (s, 1H, Ar-N**H**), 8.54 (d, *J* = 2.0 Hz, 1H, N**H**-CH), 7.97–7.88 (m, 3H, Ar-H, Ar-C**H=**N), 7.83 (d, *J* = 8.0 Hz, 2H, Ar-H), 7.48 (d, *J* = 8.0 Hz, 1H, Ar-H), 7.31 (d, *J* = 8.0 Hz, 1H), 7.05 (d, *J* = 2.0 Hz, 1H, C=C**H**-NH), 6.99 (t, *J* = 7.6 Hz, 1H, Ar-H), 6.87 (t, *J* = 7.6 Hz, 1H, Ar-H), 4.43–4.36 (m, 1H, C**H**-NH), 4.02 (d, *J* = 16.8 Hz, 1H, C**H**_2_-C=O), 3.35 (dd, *J* = 14.4, 4.8 Hz, 1H, C**H**_2_-CH), 3.14 (dd, *J* = 14.4, 4.8 Hz, 1H, C**H**_2_-CH), 3.08 (d, *J* = 16.8 Hz, 1H, C**H**_2_-C=O). ^13^C NMR (100 MHz, DMSO*-d_6_*) *δ* 164.2, 163.9, 144.3, 139.3, 136.4, 133.2, 128.4, 127.7, 125.4, 121.4, 119.2, 119.1, 118.9, 112.5, 111.7, 108.1, 56.3, 49.0, 30.9. HRMS (ESI) C_21_H_17_N_5_O_2_ calcd for (M+H)^+^ 372.1460, found 372.1453.

(*S*)-3-((1*H*-indol-3-yl)methyl)-1-(biphenyl-4-ylmethyleneamino)piperazine-2,5-dione (**12**). White solid, 0.31 g, 97%, mp 315–316 °C. ^1^H NMR (400 MHz, DMSO*-d_6_*) *δ* 10.98 (s, 1H, Ar-N**H**), 8.51 (d, *J* = 2.4 Hz, 1H, N**H**-CH), 7.94 (s, 1H, Ar-C**H=**N), 7.77 (s, 4H, Ar-H), 7.73 (d, *J* = 7.6 Hz, 2H, Ar-H), 7.53–7.45 (m, 3H, Ar-H), 7.40 (t, *J* = 7.6 Hz, 1H, Ar-H), 7.32 (d, *J* = 8.0 Hz, 1H, Ar-H), 7.06 (d, *J* = 2.0 Hz, 1H, C=C**H**-NH), 7.01 (t, *J* = 7.2 Hz, 1H, Ar-H), 6.90 (t, *J* = 7.2 Hz, 1H, Ar-H), 4.36 (d, *J* = 2.8 Hz, 1H, C**H**-NH), 4.01 (d, *J* = 16.4 Hz, 1H, C**H**_2_-C=O), 3.40–3.33 (m, 1H, C**H**_2_-CH), 3.17–3.08 (m, 2H, C**H**_2_-CH, C**H**_2_-C=O). ^13^C NMR (100 MHz, DMSO*-d_6_*) *δ* 163.7, 163.4, 146.5, 141.7, 139.3, 135.9, 133.4, 129.0, 128.1, 127.9, 127.3, 126.9, 126.7, 124.8, 120.9, 118.8, 118.4, 111.2, 107.8, 55.8, 48.7, 30.3.HRMS (ESI) C_26_H_22_N_4_O_2_ calcd for (M+H)^+^ 423.1821, found 423.1820.

(*S*)-3-((1*H*-indol-3-yl)methyl)-1-(3-chlorobenzylideneamino)piperazine-2,5-dione (**13**). White solid, 0.55 g, 93%, mp 257–258 °C. ^1^H NMR (400 MHz, DMSO*-d_6_*) *δ* 10.97 (s, 1H, Ar-N**H**), 8.53 (s, 1H, N**H**-CH), 7.87 (s, 1H, Ar-C**H=**N), 7.71 (s, 1H, Ar-H), 7.62 (d, *J* = 6.8 Hz, 1H, Ar-H), 7.52–7.43 (m, 3H, Ar-H), 7.31 (d, *J* = 8.0 Hz, 1H, Ar-H), 7.05 (s, 1H, C=C**H**-NH), 7.00 (t, *J* = 7.6 Hz, 1H, Ar-H), 6.88 (t, *J* = 7.6 Hz, 1H, Ar-H), 4.43–4.33 (m, 1H, C**H**-NH), 3.98 (d, *J* = 16.8 Hz, 1H, C**H**_2_-C=O), 3.34 (dd, *J* = 14.4, 4.4 Hz, 1H, C**H**_2_-CH), 3.13 (dd, *J* = 14.4, 4.4 Hz, 1H, C**H**_2_-CH), 3.06 (d, *J* = 16.8 Hz, 1H, C**H**_2_-C=O). ^13^C NMR (100 MHz, DMSO*-d_6_*) *δ* 164.0, 145.1, 137.0, 136.4, 134.0, 131.2, 130.3, 127.7, 127.0, 126.7, 125.3, 121.4, 119.2, 118.9, 111.7, 108.2, 30.8. HRMS (ESI) C_20_H_17_ClN_4_O_2_ calcd for (M+H)^+^ 381.1118, found 381.1110

(*S*)-3-((1*H*-indol-3-yl)methyl)-1-(2-chlorobenzylideneamino)piperazine-2,5-dione (**14**). White solid, 0.73 g, 99%, mp 195–196 °C. ^1^H NMR (400 MHz, DMSO*-d_6_*) *δ* 10.99 (s, 1H, Ar-N**H**), 8.52 (s, 1H, N**H**-CH), 8.14 (s, 1H, Ar-C**H=**N), 7.88 (d, *J* = 7.6 Hz, 1H, Ar-H), 7.52 (d, *J* = 7.6 Hz, 1H, Ar-H), 7.50–7.39 (m, 3H, Ar-H), 7.32 (d, *J* = 8.0 Hz, 1H, Ar-H), 7.06 (s, 1H, C=C**H**-NH), 6.99 (t, *J* = 7.6 Hz, 1H, Ar-H), 6.87 (t, *J* = 7.6 Hz, 1H, Ar-H), 4.40–4.34 (m, 1H, C**H**-NH), 4.02 (d, *J* = 16.4 Hz, 1H, C**H**_2_-C=O), 3.36 (dd, *J* = 14.4, 3.6 Hz, 1H, C**H**_2_-CH), 3.15–3.04 (m, 2H, C**H**_2_-C=O, C**H**_2_-CH). ^13^C NMR (100 MHz, DMSO*-d_6_*) *δ* 164.3, 164.2, 143.5, 136.4, 134.1, 132.3, 131.8, 130.4, 128.0, 127.8, 127.6, 125.5, 121.5, 119.1, 118.9, 56.3, 49.6, 30.8. HRMS (ESI) C_20_H_17_ClN_4_O_2_calcd for (M+H)^+^ 381.1118, found 381.1110.

(*S*)-3-((1*H*-indol-3-yl)methyl)-1-(2,4-dichlorobenzylideneamino)piperazine-2,5-dione (**15**). White solid, 0.44 g, 91%, mp 121–122 °C. ^1^H NMR (400 MHz, DMSO*-d_6_*) *δ* 10.99 (s, 1H, Ar-N**H**), 8.53 (s, 1H, N**H**-CH), 8.06 (s, 1H, Ar-C**H=**N), 7.88 (d, *J* = 8.4 Hz, 1H, Ar-**H**), 7.70 (s, 1H, Ar-H), 7.51 (d, *J* = 8.4 Hz, 1H, Ar-**H**), 7.45 (d, *J* = 8.0 Hz, 1H, Ar-**H**), 7.32 (d, *J* = 8.0 Hz, 1H, Ar-**H**), 7.07 (s, 1H, C=C**H**-NH), 6.99 (t, *J* = 7.2 Hz, 1H, Ar-**H**), 6.87 (t, *J* = 7.2 Hz, 1H, Ar-**H**), 4.40–4.36 (m, 1H, NH-C**H**-CH_2_), 4.03 (d, *J* = 16.4 Hz, 1H, C**H**_2_-C=O), 3.38–3.32 (m, 1H, C**H**_2_-CH), 3.17–3.02 (m, 2H, C**H**_2_-C=O, C**H**_2_-CH). ^13^C NMR (100 MHz, DMSO*-d_6_*) *δ* 164.4, 164.1, 142.0, 136.4, 135.9, 134.8, 131.0, 129.8, 129.0, 128.4, 127.6, 125.5, 121.5, 119.1, 118.9, 111.7, 108.1, 56.3, 49.5, 30.9.HRMS (ESI) C_20_H_16_Cl_2_N_4_O_2_calcd for (M+H)^+^ 415.0728, found 415.0717.

(*S*)-3-((1*H*-indol-3-yl)methyl)-1-(3,4-dichlorobenzylideneamino)piperazine-2,5-dione (**16**). White solid, 0.59 g, 92%, mp 285–286 °C. ^1^H NMR (400 MHz, DMSO*-d_6_*) *δ* 10.97 (s, 1H, Ar-N**H**), 8.54 (s, 1H, N**H**-CH), 7.86 (d, *J* = 4.8 Hz, 2H, Ar-C**H=**N, Ar-H), 7.72 (d, *J* = 8.0 Hz, 1H, Ar-H), 7.65 (d, *J* = 8.4 Hz, 1H, Ar-H), 7.47 (d, *J* = 8.0 Hz, 1H, Ar-H), 7.31 (d, *J* = 8.0 Hz, 1H, Ar-H), 7.05 (s, 1H, C=C**H**-NH), 6.99 (t, *J* = 7.6 Hz, 1H, Ar-H), 6.88 (t, *J* = 7.6 Hz, 1H, Ar-H), 4.40–4.34 (m, 1H, NH-C**H**-CH_2_), 3.97 (d, *J* = 16.4 Hz, 1H, C**H**_2_-C=O), 3.34 (dd, *J* = 14.4, 3.6 Hz, 1H, C**H**_2_-CH), 3.12 (dd, *J* = 14.4, 4.4 Hz, 1H, C**H**_2_-CH), 3.04 (d, *J* = 16.8 Hz, 1H, C**H**_2_-C=O). ^13^C NMR (100 MHz, DMSO*-d_6_*) *δ* 164.1, 164.0, 143.8, 136.4, 135.6, 132.9, 132.1, 131.6, 129.2, 127.7, 127.7, 125.4, 121.4, 119.2, 118.9, 111.7, 108.1, 56.3, 49.0, 30.9. HRMS (ESI) C_20_H_16_Cl_2_N_4_O_2_calcd for (M+H)^+^ 415.0728, found 415.0718.

(*S*)-3-((1*H*-indol-3-yl)methyl)-1-(3-nitrobenzylideneamino)piperazine-2,5-dione(**17**). White solid, 0.53 g, 89%, mp 245–246 °C. ^1^H NMR (400 MHz, DMSO*-d_6_*) *δ* 10.97 (s, 1H, Ar-N**H**), 8.54 (d, *J* = 2.4 Hz, 1H, N**H**-CH), 8.49 (s, 1H, Ar-C**H=**N), 8.26 (dd, *J* = 8.0, 1.6 Hz, 1H, Ar-H), 8.08 (d, *J* = 8.0 Hz, 1H, Ar-H), 8.00 (s, 1H, Ar-H), 7.74 (t, *J* = 8.0 Hz, 1H, Ar-H), 7.48 (d, *J* = 8.0 Hz, 1H, Ar-H), 7.31 (d, *J* = 8.0 Hz, 1H, Ar-H), 7.05 (d, *J* = 2.0 Hz, 1H, C=C**H**-NH), 6.98 (t, *J* = 7.2 Hz, 1H, Ar-H), 6.88 (t, *J* = 7.2 Hz, 1H, Ar-H), 4.42–4.35 (m, 1H, NH-C**H**-CH_2_), 4.01 (d, *J* = 16.4 Hz, 1H, C**H**_2_-C=O), 3.42–3.30 (m, 1H, C**H**_2_-CH), 3.14 (dd, *J* = 14.4, 4.8 Hz, 1H, C**H**_2_-CH), 3.07 (d, *J* = 16.4 Hz, 1H, C**H**_2_-C=O). ^13^C NMR (100 MHz, DMSO*-d_6_*) *δ* 164.2, 163.9, 148.6, 144.0, 136.6, 136.4, 134.0, 130.9, 127.7, 125.4, 124.9, 121.9, 121.4, 119.2, 118.9, 111.7, 108.1, 56.3, 49.0, 30.9. HRMS (ESI) C_20_H_17_N_5_O_4_calcd for (M+H)^+^ 392.1359, found 392.1347.

(*S*)-3-((1*H*-indol-3-yl)methyl)-1-(3-methoxybenzylideneamino)piperazine-2,5-dione (**18**). White solid, 0.53 g, 92%, mp 251–252 °C. ^1^H NMR (400 MHz, DMSO*-d_6_*) *δ* 10.98 (s, 1H, Ar-N**H**), 8.51 (s, 1H, N**H**-CH), 7.88 (s, 1H, Ar-C**H=**N), 7.49 (d, *J* = 8.0 Hz, 1H, Ar-H), 7.39–7.30 (m, 2H, Ar-H), 7.29–7.21 (m, 2H, Ar-H), 7.06 (s, 1H, C=C**H**-NH), 7.04–6.97 (m, 2H, Ar-H), 6.90 (t, *J* = 7.6 Hz, 1H, Ar-H), 4.40–4.30 (m, 1H, NH-C**H**-CH_2_), 3.98 (d, *J* = 16.8 Hz, 1H, C**H**_2_-C=O), 3.78 (s, 3H, -OC**H_3_**), 3.40–3.30 (m, 1H, C**H**_2_-CH), 3.18–3.05 (m, 2H, C**H**_2_-CH, C**H**_2_-C=O). ^13^C NMR (100 MHz, DMSO*-d_6_*) *δ* 164.2, 163.9, 159.9, 147.6, 136.4, 136.1, 130.3, 127.8, 125.3, 121.4, 120.7, 119.2, 118.9, 116.7, 112.5, 111.7, 108.3, 56.3, 55.6, 49.2, 30.8. HRMS (ESI) C_21_H_20_N_4_O_3_ calcd for (M+H)^+^ 377.1613, found 377.1611.

(*S*)-3-((1*H*-indol-3-yl)methyl)-1-(2-methoxybenzylideneamino)piperazine-2,5-dione (**19**). White solid, 0.55 g, 95%, mp 214–215 °C. ^1^H NMR (400 MHz, DMSO*-d_6_*) *δ* 10.98 (d, *J* = 1.6 Hz, 1H, Ar-N**H**), 8.48 (d, *J* = 2.4 Hz, 1H, N**H**-CH), 8.20 (s, 1H, Ar-C**H=**N), 7.77 (dd, *J* = 8.0, 1.6 Hz, 1H, Ar-H), 7.48 (d, *J* = 8.0 Hz, 1H, Ar-H), 7.46–7.41 (m, 1H, Ar-H), 7.33 (d, *J* = 8.0 Hz, 1H, Ar-H), 7.08 (d, *J* = 8.0 Hz, 1H, Ar-H), 7.06 (d, *J* = 2.4 Hz, 1H, C=C**H**-NH), 7.04–6.98 (m, 2H, Ar-H), 6.93–6.87 (m, 1H, Ar-H), 4.44–4.25 (m, 1H, NH-C**H**-CH_2_), 3.94 (d, *J* = 16.4 Hz, 1H, C**H**_2_-C=O), 3.83 (s, 3H, -OC**H_3_**), 3.34 (dd, *J* = 14.4, 4.0 Hz,1H, C**H**_2_-CH), 3.15–3.05 (m, 2H, C**H**_2_-CH, C**H**_2_-C=O). ^13^C NMR (100 MHz, DMSO*-d_6_*) *δ* 164.4, 163.9, 158.5, 144.5, 136.4, 132.6, 127.7, 126.3, 125.3, 122.4, 121.4, 121.1, 119.2, 118.9, 112.3, 111.7, 108.3, 56.4, 56.1, 49.7, 30.7. HRMS (ESI) C_21_H_20_N_4_O_3_ calcd for (M+H)^+^ 377.1613, found 377.1607.

(*S*)-3-((1*H*-indol-3-yl)methyl)-1-(3,4-dimethoxybenzylideneamino)piperazine-2,5-dione (**20**). White solid, 0.47 g, 95%, mp 255–256 °C. ^1^H NMR (400 MHz, DMSO*-d_6_*) *δ* 10.97 (s, 1H, Ar-N**H**)), 8.48 (d, *J* = 1.6 Hz, 1H, N**H**-CH), 7.88 (s, 1H, Ar-C**H=**N), 7.51 (d, *J* = 7.6 Hz, 1H, Ar-H), 7.33 (d, *J* = 8.0 Hz, 1H, Ar-H), 7.28 (d, *J* = 1.2 Hz, 1H, Ar-H), 7.20 (dd, *J* = 8.0, 1.2 Hz, 1H, Ar-H), 7.07 (d, *J* = 2.0 Hz, 1H, C=C**H**-NH), 7.02 (t, *J* = 7.6 Hz, 2H, Ar-H), 6.92 (t, *J* = 7.6 Hz, 1H, Ar-H), 4.37–4.29 (m, 1H, NH-C**H**-CH_2_), 3.95 (d, *J* = 16.4 Hz, 1H, C**H**_2_-C=O), 3.79 (d, *J* = 5.2 Hz, 6H, -OC**H_3_**, -OC**H_3_**), 3.37–3.30 (m, 1H, C**H**_2_-CH), 3.18–3.08 (m, 2H, C**H**_2_-CH, C**H**_2_-C=O). ^13^C NMR (100 MHz, DMSO*-d_6_*) *δ* 164.4, 163.6, 151.4, 149.4, 149.3, 136.4, 127.8, 127.2, 125.2, 122.7, 121.4, 119.3, 118.9, 111.8, 111.7, 109.3, 108.4, 56.3, 56.0, 55.9, 49.6, 30.7.HRMS (ESI) C_22_H_22_N_4_O_4_ calcd for (M+H)^+^ 407.1719, found 407.1713.

(*S*)-3-((1*H*-indol-3-yl)methyl)-1-(benzo [*d*] [3]dioxol-5-ylmethyleneamino)piperazine-2,5-dione (**21**). White solid, 0.45 g, 93%, mp 277–278 °C. ^1^H NMR (400 MHz, DMSO*-d_6_*) δ 10.96 (s, 1H, Ar-N**H**), 8.47 (d, *J* = 2.4 Hz, 1H, N**H**-CH), 7.85 (s, 1H, Ar-C**H=**N), 7.48 (d, *J* = 8.0 Hz, 1H, Ar-H), 7.31 (d, *J* = 8.0 Hz, 1H, Ar-H), 7.22 (d, *J* = 0.8 Hz, 1H, Ar-H), 7.14 (dd, *J* = 8.0, 1.2 Hz, 1H, Ar-H), 7.05 (d, *J* = 2.0 Hz, 1H, C=C**H**-NH), 7.03–6.95 (m, 2H, Ar-H), 6.89 (t, *J* = 7.6 Hz, 1H, Ar-H), 6.09 (s, 2H, -O-C**H_2_**-O), 4.35–4.29 (m, 1H, NH-C**H**-CH_2_), 3.93 (d, *J* = 16.4Hz, 1H, C**H**_2_-C=O), 3.38–3.26 (m, 1H, C**H**_2_-CH), 3.16–3.03 (m, 2H, C**H**_2_-CH, C**H**_2_-C=O). ^13^C NMR (100 MHz, DMSO*-d_6_*) δ 164.3, 163.7, 149.8, 148.3, 148.1, 136.4, 129.1, 127.8, 125.3, 124.5, 121.4, 119.2, 118.9, 111.7, 108.9, 108.3, 105.8, 102.1, 56.3, 49.4, 30.7. HRMS (ESI) C_21_H_18_N_4_O_4_ calcd for (M+H)^+^ 391.1406, found 391.1404.

(*S*)-3-((1*H*-indol-3-yl)methyl)-1-((2,3-dihydrobenzo [*b*] [1,4]dioxin-6-yl)methyleneamino)piperazine-2,5-dione (**22**). White solid, 0.55 g, 88%, mp 254–255 °C. ^1^H NMR (400 MHz, DMSO*-d_6_*) *δ* 10.96 (s, 1H, Ar-N**H**), 8.46 (s, 1H, N**H**-CH), 7.80 (s, 1H, Ar-C**H=**N), 7.48 (d, *J* = 8.0 Hz, 1H, Ar-H), 7.31 (d, *J* = 8.0 Hz, 1H, Ar-H), 7.20–7.14 (m, 2H, Ar-H), 7.07–6.98 (m, 2H, Ar-H, C=C**H**-NH), 6.94–6.86 (m, 2H, Ar-H), 4.32 (d, *J* = 2.4 Hz, 1H, NH-C**H**-CH_2_), 4.27 (s, 4H, -OC**H_2_**C**H_2_**-O), 3.92 (d, *J* = 16.4 Hz, 1H, C**H**_2_-C=O), 3.33–3.30 (m, 1H, C**H**_2_-CH), 3.15–3.02 (m, 2H, C**H**_2_-CH, C**H**_2_-C=O). ^13^C NMR (100 MHz, DMSO*-d_6_*) *δ* 164.3, 163.6, 148.0, 145.9, 143.9, 136.4, 128.1, 127.8, 125.3, 121.6, 121.4, 119.2, 118.9, 117.8, 116.3, 111.7, 108.3, 64.8, 64.5, 56.3, 49.3, 30.7. HRMS (ESI) C_22_H_20_N_4_O_4_ calcd for (M+H)^+^ 391.1406, found 391.1404.

(*S*)-3-((1*H*-indol-3-yl)methyl)-1-(4-bromo-2,6-difluorobenzylideneamino)piperazine-2,5-dione (**23**). Yellow solid, 0.67 g, 94%, mp= 223–224 °C. ^1^H NMR (400 MHz, DMSO*-d_6_*) δ 10.97 (s, 1H, Ar-N**H**), 8.48 (s, 1H, N**H**-CH), 7.83 (s, 1H, Ar-C**H=**N), 7.58 (d, *J* = 8.0 Hz, 2H, Ar-H), 7.44 (d, *J* = 8.0 Hz, 1H, Ar-H), 7.32 (d, *J* = 8.0 Hz, 1H, Ar-H), 7.05 (s, 1H, C=C**H**-NH), 7.01 (t, *J* = 7.6 Hz, 1H, Ar-H), 6.88 (t, *J* = 7.6 Hz, 1H, Ar-H), 4.40–4.30 (m, 1H, NH-C**H**-CH_2_), 3.99 (d, *J* = 16.4 Hz, 1H, C**H**_2_-C=O), 3.36 (m, 1H, C**H**_2_-CH), 3.19–3.06 (m, 2H, C**H**_2_-CH, C**H**_2_-C=O). ^13^C NMR (100 MHz, DMSO*-d_6_*) δ 164.3, 164.1, 161.9, 161.8, 159.3, 159.3, 137.4, 136.4, 127.7, 125.5, 123.7, 121.4, 119.1, 118.9, 116.8, 116.5, 111.7, 108.2, 56.3, 49.2, 30. HRMS (ESI) C_20_H_15_BrF_2_N_4_O_2_ calcd for (M+H)^+^ 461.0416, found461.0424.

*(S*)-3-((1*H*-indol-3-yl)methyl)-1-(3,5-di-tert-butyl-4-hydroxybenzylideneamino)piperazine-2,5-dione (**24**). Yellow solid, 0.47 g, 86%, mp 148–149 °C. ^1^H NMR (400 MHz, DMSO*-d_6_*) *δ* 10.97 (s, 1H, Ar-N**H**), 8.45 (d, *J* = 2.4 Hz, 1H, N**H**-CH), 7.82 (s, 1H, Ar-C**H=**N), 7.51 (d, *J* = 8.0 Hz, 1H, Ar-H), 7.46 (d, *J* = 5.6 Hz, 3H, Ar-H, Ar-O**H**), 7.33 (d, *J* = 8.0 Hz, 1H, Ar-H), 7.09–7.01 (m, 2H, C=C**H**-NH, Ar-H), 6.94 (t, *J* = 7.6 Hz, 1H, Ar-H), 4.33–4.24 (m, 1H, NH-C**H**-CH_2_), 3.91 (d, *J* = 16.8 Hz, 1H, C**H**_2_-C=O), 3.32 (dd, *J* = 14.4, 4.2 Hz, 1H, C**H**_2_-CH), 3.16–3.08 (m, 2H, C**H**_2_-CH, C**H**_2_-C=O), 1.40 (s, 18H, -C-(C**H_3_**)_3_, -C-(C**H_3_**)_3_). ^13^C NMR (100 MHz, DMSO*-d_6_*) *δ* 164.0, 162.7, 156.5, 152.2, 139.0, 135.9, 127.4, 125.2, 124.7, 124.6, 120.9, 118.8, 118.4, 111.3, 107.9, 55.8, 49.4, 34.5, 30.1. HRMS (ESI)C_28_H_34_N_4_O_3_calcd for (M+H)^+^ 475.2709, found 475.2703.

(*S*)-3-((1*H*-indol-3-yl)methyl)-1-(1-phenylethylideneamino)piperazine-2,5-dione (**25**). White solid, 0.65 g, 55%, mp 241–242 °C. ^1^H NMR (400 MHz, DMSO*-d_6_*) *δ* 10.98 (s, 1H, Ar-N**H**), 8.42 (s, 1H, N**H**-CH), 7.78 (d, *J* = 7.2 Hz, 2H, Ar-H), 7.55 (d, *J* = 8.0 Hz, 1H, Ar-H), 7.50 (t, *J* = 7.2 Hz, 1H, Ar-H), 7.43 (t, *J* = 7.2 Hz, 2H, Ar-H), 7.35 (d, *J* = 8.0 Hz, 1H, Ar-H), 7.11 (d, *J* = 2.0 Hz, 1H, C=C**H**-NH), 7.08 (t, *J* = 7.2 Hz, 1H, Ar-H), 6.97 (t, *J* = 7.2 Hz, 1H, Ar-H), 4.36–4.31 (m, 1H, NH-C**H**-CH_2_), 3.76 (d, *J* = 16.4 Hz, 1H, C**H**_2_-C=O), 3.41–3.29 (m, 2H, C**H**_2_-C=O, C**H**_2_-CH), 3.12 (dd, *J* = 14.8, 4.4Hz, 1H, C**H**_2_-CH), 1.91 (s, 3H, N=C-C**H_3_**). ^13^C NMR (100 MHz, DMSO*-d_6_*) *δ* 171.6, 165.0, 161.4, 136.9, 136.5, 131.4, 128.9, 128.0, 127.6, 125.1, 121.4, 119.5, 118.9, 111.7, 108.7, 56.1, 52.0, 29.8, 17.2. HRMS (ESI)C_21_H_20_N_4_O_2_ calcd for (M+H)^+^ 361.1664, found 361.1665.

(*S*)-3-((1*H*-indol-3-yl)methyl)-1-((1*H*-pyrrol-2-yl)methyleneamino)piperazine-2,5-dione (**26**). Brown solid, 0.45 g, 88%, mp 254–255 °C. ^1^H NMR (400 MHz, DMSO*-d_6_*) *δ* 11.43 (s, 1H, Pyrrole-N**H**), 10.96 (s, 1H, Ar-N**H**), 8.43 (s, 1H, N**H**-CH), 7.87 (s, 1H, Ar-C**H=**N), 7.51 (d, *J* = 7.6 Hz, 1H, Ar-H), 7.33 (d, *J* = 8.0 Hz, 1H, Ar-H), 7.08–7.01 (m, 2H, C=C**H**-NH, Ar-H), 6.96–6.87 (m, 2H, Ar-H, Pyllore-H), 6.45 (s, 1H, Pyllore-H), 6.13 (s, 1H, Pyllore-H), 4.33–4.26 (m, 1H, NH-C**H**-CH_2_), 3.84 (d, *J* = 16.8 Hz, 1H, C**H**_2_-C=O), 3.37–3.29 (m, 1H, C**H**_2_-CH), 3.14–3.05 (m, 2H, C**H**_2_-C=O, C**H**_2_-CH). ^13^C NMR (100 MHz, DMSO*-d_6_*) *δ* 164.6, 163.2, 145.6, 136.4, 127.9, 127.0, 125.2, 123.6, 121.5, 119.2, 119.0, 115.6, 111.7, 109.8, 108.4, 56.4, 50.4, 30.5. HRMS (ESI) C_18_H_17_N_5_O_2_calcd for (M+H)^+^ 336.1460, found 336.1458.

(*S*)-3-((1*H*-indol-3-yl)methyl)-1-(thiophen-2-ylmethyleneamino)piperazine-2,5-dione (**27**). Yellow solid 0.53 g, 97%, mp 258–259 °C. ^1^H NMR (400 MHz, DMSO*-d_6_*) *δ* 10.96 (s, 1H, Ar-N**H**), 8.47 (s, 1H, N**H**-CH), 8.19 (s, 1H, Ar-C**H=**N), 7.67 (d, *J* = 4.0 Hz, 1H, Thiophene-H), 7.48 (d, *J* = 7.6 Hz, 1H, Ar-H), 7.40 (d, *J* = 4.0 Hz, 1H, Thiophene-H), 7.32 (d, *J* = 8.0 Hz, 1H, Ar-H), 7.12 (t, 1H, Thiophene-H), 7.08–6.97 (m, 2H, C=C**H**-NH, Ar-H), 6.91 (t, *J* = 7.2 Hz, 1H, Ar-H), 4.40–4.25 (m, 1H, NH-C**H**-CH_2_), 3.93 (d, *J* = 16.4 Hz, 1H, C**H**_2_-C=O), 3.41–3.03 (m, 3H, C**H**_2_-C=O, C**H_2_**-CH). ^13^C NMR (100 MHz, DMSO*-d_6_*) *δ* 164.2, 163.7, 143.9, 139.4, 136.4, 132.5, 129.9, 128.3, 127.8, 125.3, 121.4, 119.2, 118.9, 111.7, 108.3, 56.3, 49.6, 30.7. HRMS (ESI) C_18_H_16_N_4_O_2_S calcd for (M+H)^+^ 353.1072, found 353.1063.

(*S*)-3-((1*H*-indol-3-yl)methyl)-1-(furan-2-ylmethyleneamino)piperazine-2,5-dione (**28**). Yellow solid, 0.58 g, 88%, mp 243–244 °C. ^1^H NMR (400 MHz, DMSO*-d_6_*) *δ* 10.96 (s, 1H, Ar-N**H)**, 8.48 (s, 1H, N**H**-CH), 7.85 (d, *J* = 8.4 Hz, 2H), 7.47 (d, *J* = 7.6 Hz, 1H), 7.31 (d, *J* = 8.0 Hz, 1H), 7.09–6.76 (m, 4H), 6.61 (s, 1H), 4.40–4.24(m, 1H, NH-C**H**-CH_2_), 3.91 (d, *J* = 16.4 Hz, 1H, C**H**_2_-C=O), 3.38–3.25 (m, 1H, C**H_2_**-CH), 3.15–2.97 (m, 2H, C**H**_2_-C=O, C**H_2_**-CH). ^13^C NMR (100 MHz, DMSO*-d_6_*) *δ* 164.1, 163.8, 149.8, 145.9, 138.2, 136.4, 127.8, 125.3, 121.4, 119.2, 118.9, 115.5, 112.7, 111.7, 108.2, 56.3, 49.4, 30.7. HRMS (ESI) C_18_H_16_N_4_O_3_calcd for (M+H)^+^ 337.1300, found 337.1293.

(*S*)-1-((1*H*-imidazol-2-yl)methyleneamino)-3-((1*H*-indol-3-yl)methyl)piperazine-2,5-dione (**29**). White solid, 0.35 g, 54%, mp 172–173 °C. ^1^H NMR (400 MHz, DMSO*-d_6_*) *δ* 12.68 (s, 1H, Imidazole-N**H**), 10.98 (s, 1H, Ar-N**H**), 8.50 (d, *J* = 2.4 Hz, 1H, N**H**-CH), 7.83 (s, 1H, Ar-C**H=**N), 7.48 (d, *J* = 7.6 Hz, 1H, Ar-H), 7.32 (d, *J* = 8.0 Hz, 1H, Ar-H), 7.15 (s, 2H, Imidazole-**H**),7.04 (d, *J* = 2.0 Hz, 1H, C=C**H**-NH), 7.01 (t, *J* = 7.6 Hz, 1H, Ar-H), 6.89 (t, *J* = 7.6 Hz, 1H, Ar-H), 4.56–4.19 (m, 1H, NH-C**H**-CH_2_), 3.98 (d, *J* = 16.8 Hz, 1H, C**H**_2_-C=O), 3.33 (dd, *J* = 14.8, 4.0 Hz, 1H, C**H_2_**-CH), 3.17–3.01 (m, 2H, C**H**_2_-C=O, C**H_2_**-CH). ^13^C NMR (100 MHz, DMSO*-d_6_*) *δ* 163.7, 163.5, 142.3, 139.6, 135.9, 127.3, 124.8, 121.0, 118.6, 118.4, 111.2, 107.8, 55.9, 49.1, 30.1. HRMS (ESI) C_17_H_16_N_6_O_2_calcd for (M+H)^+^ 337.1413, found 337.1406.

(*S*)-3-((1*H*-indol-3-yl)methyl)-1-(pyridin-3-ylmethyleneamino)piperazine-2,5-dione (**30**). White solid, 0.50 g, 93%, mp 291–292 °C. ^1^H NMR (400 MHz, DMSO*-d_6_*) *δ* 10.96 (s, 1H, Ar-N**H**), 8.79 (s, 1H, 2-**H**-Pyridine), 8.60 (d, *J* = 4.4 Hz, 1H, N**H**-CH), 8.51 (s, 1H,), 8.06 (d, *J* = 7.6 Hz, 1H, Pyridine-H), 7.95 (s, 1H, Ar-C**H=**N), 7.47 (t, *J* = 7.6 Hz, 2H, Ar-H, Pyridine-H), 7.31 (d, *J* = 8.0 Hz, 1H, Ar-H), 7.06 (s, 1H, C=C**H**-NH), 6.99 (t, *J* = 7.6 Hz, 1H, Ar-H), 6.88 (t, *J* = 7.6 Hz, 1H, Ar-H), 4.41–4.32 (m, 1H, NH-C**H**-CH_2_), 4.01 (d, *J* = 16.8 Hz, 1H, C**H**_2_-C=O), 3.36 (dd, *J* = 14.0, 4.0 Hz, 1H, C**H_2_**-CH), 3.17–3.06 (m, 2H, C**H**_2_-C=O, C**H_2_**-CH). ^13^C NMR (100 MHz, DMSO*-d_6_*) *δ* 164.1, 164.0, 151.4, 149.6, 144.4, 136.4, 134.2, 130.7, 127.7, 125.4, 124.4, 121.4, 119.2, 118.9, 111.7, 108.2, 56.3, 49.1, 30.8.HRMS (ESI) C_19_H_17_N_5_O_2_calcd for (M+H)^+^ 348.1460, found 348.1455.

(*S*)-3-((1*H*-indol-3-yl)methyl)-1-(2,2-dimethylpropylideneamino)piperazine-2,5-dione (**31**). Yellow solid, 0.36 g, 71%, mp 236–237 °C. ^1^H NMR (400 MHz, DMSO*-d_6_*) *δ* 10.94 (s, 1H, Ar-N**H**), 8.34 (s, 1H, N**H**-CH), 7.41 (d, *J* = 8.0 Hz, 1H, Ar-H), 7.32 (d, *J* = 8.0 Hz, 1H, Ar-H), 7.07–6.98 (m, 3H, Ar-H, C=C**H**-NH, Ar-C**H=**N), 6.91 (t, *J* = 7.6 Hz, 1H, Ar-H), 4.26–4.17 (m, 1H, NH-C**H**-CH_2_), 3.67 (d, *J* = 16.4 Hz, 1H, C**H**_2_-C=O), 3.30–3.15 (m, 1H, C**H**_2_-CH), 3.09–3.00 (m, 1H, C**H**_2_-CH), 2.76 (d, *J* = 16.4 Hz, 1H, C**H**_2_-C=O), 0.98 (s, 9H, -C-(C**H**_3_)_3_). ^13^C NMR (100 MHz, DMSO*-d_6_*) *δ* 164.7, 163.6, 161.8, 136.4, 127.6, 125.5, 121.4, 119.3, 118.9, 111.7, 108.2, 56.2, 49.4, 35.4, 30.8, 27.5. HRMS (ESI) C_18_H_22_N_4_O_2_calcd for (M+H)^+^ 327.1821, found 327.1821.

(*S*)-3-((1*H*-indol-3-yl)methyl)-1-(cyclohexylmethyleneamino)piperazine-2,5-dione (**32**). Yellow solid, 0.41 g, 75%, mp 257–258 °C. ^1^H NMR (400 MHz, DMSO*-d_6_*) *δ* 10.94 (s, 1H, Ar-N**H**), 8.35 (s, 1H, N**H**-CH), 7.41 (d, *J* = 7.6 Hz, 1H, Ar-H), 7.31 (d, *J* = 8.0 Hz, 1H, Ar-H), 7.05–6.99 (m, 3H, Ar-C**H=**N, C=C**H**-NH, Ar-H), 6.90 (t, *J* = 7.6 Hz, 1H, Ar-H), 4.27–4.18 (m, 1H, NH-C**H**-CH_2_), 3.67 (d, *J* = 16.4 Hz, 1H, C**H**_2_-C=O), 3.32–3.24 (m, 1H, C**H**_2_-CH), 3.04 (dd, *J* = 14.4, 4.0 Hz, 1H, C**H**_2_-CH), 2.77 (d, *J* = 16.4 Hz, 1H, C**H**_2_-C=O), 2.19–2.09 **(**m, 1H, CH_2_-C**H**-CH_2_), 1.74–1.55 (m, 5H), 1.31–1.08 (m, 5H). ^13^C NMR (100 MHz, DMSO*-d_6_*) *δ* 164.6, 163.6, 158.0, 136.4, 127.7, 125.5, 121.4, 119.2, 118.9, 111.7, 108.2, 56.2, 49.2, 30.2, 30.0, 29.9, 26.0, 25.3. HRMS (ESI) C_20_H_24_N_4_O_2_calcd for (M+H)^+^ 353.1977, found 353.1976.

### 3.3. Biological Assay

The anti-TMV, larvicidal, and fungicidal activities of the synthesized compounds were tested using reported methods [24,25,26], which are described in detail in the Appendix A. Each bioassay was repeated three times; the results are presented as means ± standard errors.

## 4. Conclusions

In summary, we designed and synthesized a series of novel tryptophan derivatives containing 2,5-diketopiperazine and acyl hydrazine moieties. We systematically bioassayed the synthesized compounds and found that they possessed moderate to good activities against TMV. Compounds **4**, **9**, **14**, **19**, and **24** showed higher antiviral activity inactivation, curative, and protection activities in vivo than that of ribavirin and comparable to that of ningnanmycin. Most of the compounds exhibited broad-spectrum activity when tested against 13 kinds of phytopathogenic fungi, and they showed selective good fungicidal activities against *A. solani*, *P. capsica*, and *S. sclerotiorum*. In addition, most of these compounds were also active against *T. cinnabarinus*, *P. xylostella*, *C. pipiens pallens*, *M. separata*, *H. armigera*, and *P. nubilalis*. Further studies aimed at the optimization of the structures and elucidation of the mode of action are in progress in our laboratory.

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
