# Peer review of "Design, Synthesis, and Bioactivities of Novel Tryptophan Derivatives Containing 2,5-Diketopiperazine and Acyl Hydrazine Moieties"

_molecules, 2022, doi:10.3390/molecules27185758_

Round 1

Reviewer 1 Report

Materials and methods should be described in a specific point. 

Author Response

Dear professor:

Thank you for evaluating our manuscript. All the revision requests were responded point by point and listed below.

Materials and methods should be described in a specific point. 

Reply: Thanks for your good suggestion! The “3.1. Instruments” has been changed to “3.1. Materials”. “Ribavirin (Topscience Co.), ningnanmycin (Alta Scientific Co.), chlorothalonil (Bail-ing Agrochemical Co.), rotenone (Accela ChemBio Inc), avermectin (Chemieliva Pharma-ceutical Co.), and other reagents were purchased from commercial sources and were used as received.” has been moved from the “3.2. General Synthesis” to the “3.1. Materials”

Author Response

Dear professor:

Thank you for evaluating our manuscript. All the revision requests were responded point by point and listed below.

1. The author needs to mention the full name of TMV (tobacco mosaic virus) atlases once at the start of the introduction part.

Reply: Thanks for your good suggestion! We have added it in the introduction part.

2. The sentence in line 61,62 and 63 is very long and confusing. Please rephrase it.

Reply: Thanks for your good suggestion! The corresponding description has been changed to “In this work, to improve the anti-virus activity of tryptophan, we designed, and synthesized a series of novel tryptophan derivatives containing diketopiperazine (DKP) and acyl hydrazon moieties, and first evaluated their biological activities.”

3. The chemical formula of Chloro acetyl chloride in scheme 1 needs to be corrected.

Reply: Thanks for your good suggestion! We have corrected it.

4. The author should describe the brief synthesis of the compounds in the results and discussion part with proper references.

Reply: Thanks for your good suggestion! The brief synthesis of the compounds has been provided as follows: “Using natural amino acid L-tryptophan as raw material, through esterification, amidation, cyclization, and condensation reactions, we could easily realize the synthesis of target compounds 3-32 (Scheme 1)[23].” The reference [23] has been added.

5. Please describe the characterization of the final compounds with the salient features of the final compound in the results and discussion section.

Reply: Thanks for your good suggestion! The corresponding description has been added in the results and discussion part: “Compared with compound 3, the notable feature of the 1H NMR spectrum of these target compounds was an additional single peak of imine hydrogen (See supporting information for details).”

6. In the supplementary information, in all 1H NMR spectrums, proton integration merged into the δ scale of the spectrum. Please take care of it.

Reply: Thanks for your good suggestion! We have modified all the 1H NMR spectrums.

7. What solvent system was used to purify the final compounds?

Reply: Thanks for your question! Dichloromethane: methanol = 10:1 was used as the eluent, Related information has been added to “3.2. General Synthesis”

8. Line 97, page 5, replaces the word ‘substituted’ with ‘substituents’.

Reply: Thanks for your good suggestion! We have corrected it.

9. SAR is poorly written; need to modify it and make it more informative. For example, in lines, 101 and 102 authors mentioned CN is a weak electron withdrawing group which is not in reality. CN group is a very strong electron-withdrawing group. Please rewrite this sentence.

Reply: Thanks for your good suggestion! The corresponding description has been changed to “For the substituents at the para position of the benzene ring, electron-donating groups (6, 9) and weak electron-withdrawing group (8) were favorable for maintaining the activity.”

10. The fungicidal spectrum of most acyl hydrazone derivatives (4-32) was wider than compound 3. (Line 143) What is the meaning of the word ‘wider’. Need to write more scientific word.

Reply: Thanks for your good suggestion! ‘wider’ has been changed to “broader”.

Reviewer 3 Report

Wang and co-workers designed and synthesized a series of tryptophan derivatives containing 2,5-diketopiperazine and acyl hydrazine moieties. Compounds have been tested for antiviral activities against tobacco mosaic virus. Manuscript is well-written and summarized. Here are my suggestions to improve the quality of manuscript.

1. Few typo errors are there. e.g. lavicidal in abstract. Please check the whole draft.

2. Plant names should be italic.

3. Add solvent system for TLC (reaction progress).

4. Be consistent in presentation e.g. page 10; 91 %, 95 % and 80%.

5. The introduction needs more literature citation. Old references are cited. Add recent literature. 

Author Response

Dear professor:

Thank you for evaluating our manuscript. All the revision requests were responded point by point and listed below.

1. Few typo errors are there. e.g. lavicidal in abstract. Please check the whole draft.

Reply: Thanks for your good suggestion! We have corrected it.

2. Plant names should be italic.

Reply: Thanks for your good suggestion! All the names have been changed to italic in the notes to Table 3.

3. Add solvent system for TLC (reaction progress).

Reply: Thanks for your good suggestion! Dichloromethane: methanol = 10:1 was used as the eluent, Related information has been added to “3.2. General Synthesis”

4. Be consistent in presentation e.g. page 10; 91 %, 95 % and 80%.

Reply: Thanks for your good suggestion! We have corrected them.

5. The introduction needs more literature citation. Old references are cited. Add recent literature. 

Reply: Thanks for your good suggestion! New references [1],[5],[6],[8], and [10] have been added.

Round 2

Reviewer 1 Report

Accept.